# Expression of POU2F3 Transcription Factor Control Inflammation, Immunological Recruitment and Metastasis of Pancreatic Cancer in Mice

**DOI:** 10.3390/biology9100341

**Published:** 2020-10-19

**Authors:** Jennifer Bintz, Analía Meilerman Abuelafia, François Gerbe, Elodie Baudoin, Nathalie Auphan-Anezin, Emmanuelle Sidot, Philippe Jay, Juan Iovanna

**Affiliations:** 1Centre de Recherche en Cancérologie de Marseille (CRCM), INSERM U1068, CNRS UMR 7258, Aix-Marseille Université and Institut Paoli-Calmettes, Parc Scientifique et Technologique de Luminy, 13288 Marseille, France or jennifer.bintz@bric.ku.dk (J.B.); meilerman.analia@gmail.com (A.M.A.); 2Institute of Functional Genomics (IGF), University of Montpellier, CNRS, INSERM, Equipe Labellisee Ligue Contre le Cancer, 34000 Montpellier, France; Francois.Gerbe@igf.cnrs.fr (F.G.); emmanuelle.sidot@kennedy.ox.ac.uk (E.S.); Philippe.Jay@igf.cnrs.fr (P.J.); 3Centre d’Immunologie de Marseille-Luminy, Aix-Marseille Université, INSERM U1104, CNRS UMR 7280, Parc Scientifique et Technologique de Luminy, 13288 Marseille, France; elodie.baudoin@inserm.fr (E.B.); auphan@ciml.univ-mrs.fr (N.A.-A.)

**Keywords:** TUFT cells, POU2F3, PDAC, epithelial-to-mesenchymal transition, metastasis

## Abstract

**Simple Summary:**

The presence and the role of TUFT cells in pancreatic ductal adenocarcinoma (PDAC) is discussed. Therefore, we decided to inactivate the *POU2F3* gene, which is essential for TUFT cells development, in an aggressive PDAC mice model known as PDX1-Cre;LSL-Kras^G12D^;Ink4a^fl/fl^. Morphological and molecular analysis of *POU2F3*-deleted PDAC show not significant changes in tumors growth and survival of animals although it promotes EMT. Remarkably, we observed that in *POU2F3*-deleted animals the lack of TUFT cells prevents metastasis formation and strongly modifies the immunological and inflammatory landscape.

**Abstract:**

TUFT cells have been described as strong modulators of inflammatory cells in several tissues including pancreas. TUFT cells, also known as DCLK1^+^ cells, are dependent of the transcriptional factor POU2F3. Several works report DCLK1^+^ cells in early stages of PDAC development suggesting an important role of TUFT cells in PDAC development. Therefore, we developed a mice model (PDX1-Cre;Kras^G12D^;Ink4a^fl/fl^), known as PKI model, deficient or not of POU2F3. In this animal model, deficiency of POU2F3 results in the absence of TUFT cells in PDAC as expected. Although, tumor development and growth are not significantly influenced, the development of liver metastasis was almost completely inhibited in POU2F3-deficient mice. Surprisingly, the absence of metastasis was associated with a higher expression of epithelial-to-mesenchymal transition markers, but to a lower inflammatory microenvironment suggesting that inflammation influences metastasis production more than epithelial-to-mesenchymal transition in this animal model. We can conclude that POU2F3 could be a new therapeutic target for control PDAC progression.

## 1. Introduction

Pancreatic Ductal Adenocarcinoma (PDAC) is the fourth leading cause of death by cancer in western countries, while it is only the tenth in term of incidence according to the World Health Organisation. Those data illustrate how far we are from understanding the mechanisms of pancreatic cancer development and progression. At present, physicians are still unable to prevent it or even to propose an efficient treatment at the moment of its diagnosis. Indeed, alongside the absence of symptoms [1], chemo-resistance is another issue we have to face [2]. Moreover, with the rise of immunotherapies, new hopes are coming out and scientists attempt to detail the immune response against cancers in order to adapt and refine therapeutically strategies [3,4]. Unfortunately, these new therapies showing remarkable results in several cancers [5,6,7], are almost inefficient for treating patients with a PDAC nowadays. One of the most important reasons for the poor clinical evolution of patients suffering a PDAC is its very early metastatic capacity that kills patients [8]. Several factors are known to increase the metastatic capacity of cancer cells such as epithelial-to-mesenchymal transition [9], chemotactic factors [10], cytokines [11], adhesion molecules [12], and expression of some metalloproteinases [13], among other important factors. Another lesser-studied mechanism, likely involved, in metastasis of PDAC is inflammation [14,15]. Inflammation of PDAC is largely considered to promote PDAC and controls its aggressiveness. Although, the mechanism starting [16,17] and maintaining inflammation in tumors remain poorly described.

TUFT cells have been described as strong modulators of inflammatory cells in several tissues [18,19] including pancreas [20,21]. TUFT cells, also known as DCLK1+ cells [22], are dependent of the transcriptional factor POU2F3 [23,24]. In this way several papers report DCLK1+ cells in early stages of PDAC development such as ADM and PanIN [25,26,27,28,29] suggesting an important role of TUFT cells in PDAC development through inflammation. Therefore, we developed a mice model (PDX1-Cre;KrasG12D;Ink4aflox/flox), known as PKI model [30], in which an aggressive PDAC is genetically induced, deficient or not for POU2F3. In this model, deficiency of POU2F3 results in the absence of TUFT cells in PDAC as expected. Although, tumor growth is not significantly modified, development of liver metastasis was almost inhibited in POU2F3 deficient mice. Surprisingly, absence of metastasis was associated to a higher expression of epithelial-to-mesenchymal transition markers but to a lower inflammatory microenvironment suggesting that inflammation influences metastasis production more than epithelial-to-mesenchymal transition in this animal model. Therefore, POU2F3 could be a new therapeutic target for controlling PDAC progression.

## 2. Materials and Methods

### 2.1. Animals

The mouse model of pancreatic adenocarcinoma used in this study resulted from crossbreeding of the following strains: Pdx1-cre [31], LSL-KrasG12D6 and Ink4a/Arffl/fl [32,33]. POU2F3KO mice were previously reported [23]. The mice were kept in the Experimental Animal House of the Centre de Cancérologie de Marseille (CRCM) pole Luminy, following institutional guidelines.

### 2.2. Immunohistochemistry

Tissue dissection, fixation, and immunohistochemistry on thin sections of paraffin-embedded tissue were performed essentially as described previously. The antibodies used in this study were presented in Appendix A. Slides were washed twice times with 0.1% PBS-Tween (Sigma-Aldrich, Saint-Quentin Fallavier, France) before incubation with a secondary biotinylated antibody. Signals were developed with DAB (Sigma-Aldrich) and a haematoxylin counterstain (DiaPath) was used. After dehydration, sections were mounted in Pertex (Histolab). All experiments were performed on formalin-fixed tissues and 10 mM sodium citrate (pH 6.4) or Tris-EDTA (pH 9.0) treatment was used. Every staining was repeated in 3 different experiments and performed upon 6 individual tumors.

### 2.3. RT-qPCR

RNAeasy Mini Kit (QIAGEN) has been used for the extraction and the purification of RNA from frozen tumors. Homogenization was performed right after adding the protein precipitation buffer with iron beads thanks to the Precellys Evolution device (OZYME). Supernatant from the 10.000 RPM centrifugation for 10 min was transferred into the column for purification. Then, we went through the Reverse Transcription (GoScript PROMEGA) from 1 µg of RNA in order to get 20 µL of cDNA diluted in 200 µL of water before using 2 µL into the qPCR Master Mix (GoTaq PROMEGA). Primers were as follow in Appendix A.

### 2.4. Immunophenotyping

Warmed PBS (1×, 3 mM EDTA) has been injected into peritoneal cavity in order to aspirate the circulating cells. After centrifugation, the pellet is resuspended with a specific FACS buffer (1× PBS, 2% FCS, 1 mM EDTA, 0.1% sodium azyde). Organs were collected from mice and saved in DMEM medium (Life Technologies, Villebon sur Yvette, France) completed with 10% FCS, Penicilline, Streptomycin, Glutamine, Pyruvate, Hepes and βME. For homogenization, tumors were first manually dissociated with scissors then with GentleMACS Dissociator (Milteny Biotec, Paris, France) according to the manufacturer’s protocol. The spleen was smashed with a syringe and filtrated through nylon tissue. Counting through Malassez cell helped us to normalize every sample before the incubation with antibodies listed in the Appendix A. Analysis has been performed with a flow cytometer (BD LSR Fortessa ×20). Antibodies are mixed into blocking solution (24G2) before the incubation with the cells during 30 min at 4 °C then washed with FACS buffer. After centrifugation, cells are resuspended with PBS 1× with a viability marker then filtrated before Cytometer analysis.

### 2.5. Study Approval

All experimental protocols were carried out in accordance with the nationally approved guidelines for the treatment of laboratory animals. All experimental procedures on animals were approved by the Comité d’éthique de Marseille numéro 14 (n° 2018092011266070).

### 2.6. Statistical Analysis

Survival curves have been processed with PRISM software using the Kaplan-Meyer equation. Transcriptomic results are presented as the mean from 6 individual tumors in 3 independent experiments and have been tested according to the Mann-Whitney equation. The IHC sections come from the same individuals. Immunophenotyping have been performed with 3 independent individuals for each groups (POU2F3^−/−^ and POU2F3^+/+^) and tested with PRISM. Significance is evaluated with 1, 2 or 3 stars for *p*-value below 0.05, 0.01 and 0.001 respectively.

## 3. Results

### 3.1. TUFT Cells Are Not Essential for Pancreatic Carcinogenesis

We evaluated the role of the TUFT cells in carcinogenesis of pancreatic cancer. To this end we developed an engineered mice (PDX1-Cre;KrasG12D;Ink4aflox/flox), known as PKI model, which develop an aggressive PDAC, deficient or not for POU2F3, a transcriptional factor which is known to control the development of TUFT cells. Specific staining of the POU2F3 protein was found in the nucleus of 6.80 ± 2.12% of PDAC cells of PKI;POU2F3^+/+^ but not of PKI;POU2F3^−/−^ mice (Figure 1B). Moreover, DCLK1 protein expression was strongly detected in 8.25 ± 3.20% of PDAC cells from PKI;POU2F3^+/+^ but not in the pancreas from PKI;POU2F3^−/−^ mice (Figure 1B). These results validate that TUFT cells development are dependent of the POU2F3 expression in pancreatic cancer like in intestine [23] and lung [24], but more importantly, they show that these cells were dispensable for PDAC development. We systematically euthanized these mice at the time presenting the first signs of agony or suffering (around 9 weeks olds). Tumors were immediately removed and weighed after sacrifice from both PKI;POU2F3^+/+^ and PKI;POU2F3^−/−^ mice. Tumors showed not differences in their weigh (68 ± 4.32 mg vs. 76 ± 5.10 mg; *p*-value = 0.1) as shown in Figure 1A. The time of their sacrifice was respectively 8 ± 0.88 weeks and 9 ± 0.22 weeks olds for PKI;POU2F3^+/+^ and PKI;POU2F3^−/−^ mice (Figure 1A).

### 3.2. POU2F3 Inactivation Decreases Liver Metastasis Development

Although, PDAC growth is similar in PKI;POU2F3^+/+^ and PKI;POU2F3^−/−^ mice, we measured the metastatic capacity of these tumors to spread in liver. We obtained the livers of these animals and after the H&E staining we estimated the number of microscopic metastasis. To our surprise almost not metastasis were found in livers from PKI;POU2F3^−/−^ mice. Livers from PKI;POU2F3^+/+^ mice showed 4.2 ± 1.5 metastasis per liver whereas livers from PKI;POU2F3^−/−^ mice showed only 1.1 ± 0.6 as presented in Figure 1C. Interestingly, morphological characteristics of both PKI;POU2F3^+/+^ and PKI;POU2F3^−/−^ metastasis are similar in size and histology. We conclude that although POU2F3 deficiency does not influences tumor growth, it could play an essential role in metastasis development.

### 3.3. POU2F3 Inactivation Enhances Epithelial-to-Mesenchymal Transition in PDAC

Then, we wanted to characterize the histology of the PDAC from both PKI;POU2F3^+/+^ and PKI;POU2F3^−/−^ mice. Thus, after tumors staining with H&E we found PDAC presenting several large cystic-like structures with a monolayer of epithelial cells (Figure 1C) in PKI;POU2F3^+/+^ mice, previously described as high grade PanIN, and expressing abundant E-cadherin, also known as Cdh1 (Figure 2A). Surprisingly, those structures are not obvious in PKI;POU2F3^−/−^ mice as shown in Figure 1C. On the contrary, in the PDAC from PKI;POU2F3^−/−^ mice we are still able to describe every stage of PDAC development, with large cytoplasm and lack of nuclear polarization until the lumen invasion, associated with a very low expression of the E-cadherin (Figure 1C and Figure 2A), suggesting a different evolution of the disease in POU2F3-deficient mice, potentially linked to an increased epithelial-to-mesenchymal transition.

To address this hypothesis we measured the expression of several transcripts associated to the epithelial-to-mesenchymal transition by RT-qPCR as presented in Figure 2B. mRNA expression was calculated as relative to the 36B4 mRNA, a housekeeping ribosomal protein gene. Epithelial-associated markers such as Muc1 (16.53 ± 4.39 vs. 4.84 ± 1.97; *p*-value = 0.002) and Cdh1 (4.89 ± 3.25 vs. 2.23 ± 1.08; *p*-value = 0.004) were downregulated in PKI;POU2F3^−/−^ PDAC compared to PKI;POU2F3^+/+^ tumors. On the contrary, mesenchymal-associated markers like VIM (0.22 ± 0.08 vs. 0.59 ± 0.31; *p*-value = 0.007) and Cdh2, also known as N-cadherin (0.42 ± 0.14 vs. 0.64 ± 0.28; *p*-value = 0.03) were overexpressed in PKI;POU2F3^−/−^ PDAC compared to PKI;POU2F3^+/+^ tumors. Most importantly, GATA6, which is a marker of pancreatic differentiation and the main regulator of the classical PDAC subtype [34,35], was found 5 times higher in PDAC from PKI;POU2F3^+/+^ (23.39 ± 6.83 vs. 5.12 ± 3.02; *p*-value = 0.004) as presented in Figure 2B, indicating that PDAC cells are less differentiated in POU2F3-deficient mice.

We also measured the expression of hyaluronic receptor CD44 [36] and metalloproteinase MMP19 [37] transcripts since they are associated to the metastatic potential of the PDAC. One is making bounds with the glycosaminoglycan while the other is degrading the proteins leading the way through the extracellular matrix. We found that these markers were overexpressed in PKI;POU2F3^+/+^ PDAC compared to PKI;POU2F3^−/−^ tumors. CD44 and MMP19 expression were respectively 4.61 ± 1.76 vs. 1.94 ± 0.92; *p*-value = 0.008 and 4.65 ± 1.64 vs. 1.64 ± 0.94; *p*-value = 0.005 (Figure 2B).

Altogether, these data strongly suggest that genetic inactivation of POU2F3 improves the epithelial-to-mesenchymal transition. Surprisingly, epithelial-to-mesenchymal transition is reported to be associated to the metastatic potential of the transformed cells. However, this process is higher in PKI;POU2F3^−/−^ tumors, they developed fewer liver metastases. Therefore, we can conclude that the enhanced epithelial-to-mesenchymal transition occurring in PKI;POU2F3^−/−^ does not contribute to metastasis development.

### 3.4. POU2F3 Inactivation Inhibits the Recruitment Inflammatory Mediators into the PDAC

Since POU2F3 is an essential transcriptional factor for TUFT cells development and TUFT cells seems to play a major role in recruitment of inflammatory cells through the secretion of immunological factors, we studied the effect of POU2F3 deficiency on inflammation and immune cells recruitment in both PKI;POU2F3^+/+^ and PKI;POU2F3^−/−^ tumors. We observed the mRNA expression analyzed by RT-qPCR (Figure 3) or protein expression by immunohistochemistry (Figure 4) of several cytokines, chemokines and some of their receptors.

Histological staining of CD117-expressing cells also known as mast cells and associated with poor prognosis [38] shows a lower infiltration in PKI;POU2F3^−/−^ tumors for the benefits of CD4+ T-cells alongside CD8a-expressing cells that are more numerous regarding PKI;POU2F3^+/+^ sections. Among the stroma, we are actually counting numerous CD117-expressing cells in PKI;POU2F3^+/+^ sections regarding the PKI;POU2F3^−/−^ tissue (34.6 ± 12.6 vs. 11.3 ± 2.7 cells; *p*-value = 0.01). In the opposite, we see a stronger staining targeting CD8a and CD4 in PKI;POU2F3^−/−^, mostly around the PanIN which are not that obvious in PKI;POU2F3^+/+^ tumors, (respectively 12.2 ± 3.2 vs. 61.0 ± 12.6 cells; *p*-value = 0.06 and 18.7 ± 6.1 vs. 31.0 ± 7.9 cells; *p*-value = 0.05) as shown in Figure 4. Quantitative analysis of tumor RNA regarding specific cytokines for mast cells recruitment such as IL4 and IL9 confirmed our observations. Relative expression shown in Figure 4 is widely inferior in PKI;POU2F3^−/−^ than PKI;POU2F3^+/+^ tissues (respectively 1.25 ± 0.27 vs. 3.68 ± 1.58; *p*-value = 0.008 and 1.22 ± 0.51 vs. 10.12 ± 4.93; *p*-value = 0.01), just like IL13 (0.90 ± 0.14 vs. 14.21 ± 5.40; *p*-value = 0.005) which is known to be stimulated by the TUFT cells [23] and determinant in the macrophages polarization [18]. Indeed, RNA analysis show higher expression of interleukin IL17A and IL1b (respectively 0.89 ± 0.37 vs. 4.35 ± 0.61; *p*-value = 0.002 and 0.94 ± 0.08 vs. 4.28 ± 1.86; *p*-value = 0.001) alongside the specific cytokines of inflammation IL10, IFNg and TNFa (respectively 0.42 ± 0.13 vs. 2.08 ± 1.40; *p*-value = 0.03 and 1.43 ± 0.92 vs. 4.70 ± 2.57; *p*-value = 0.002 and 6.55 ± 1.09 vs. 11.29 ± 2.51; *p*-value = 0.01). All of those factors decreased when POU2F3 is knocked-out meaning that the inflammatory response is significantly impacted when TUFT cells are depleted.

Immuno-staining with anti-CD11b and anti-IL17R suggest that some leukocytes have been recruited on the tumoral site preferentially when POU2F3 is expressed (Figure 3). IL-17 is a proinflammatory cytokine that regulates both granulopoiesis and recruitment of neutrophils into sites of inflammation. This is due, in part, to the ability of IL-17A pathway to induce the release of CXC chemokines as well as to regulate the expression of G-CSF, a critical granulopoietic growth factor. Its receptor is known to play a pathogenic role in many inflammatory and autoimmune diseases [39] and seems to be expressed by the PanIN cells with basolateral localization in the PKI;POU2F3^+/+^ tumors. Well, this is not observed in the PKI;POU2F3^−/−^ tumors. CD11b-expressing cells are definitely more present in the PKI;POU2F3^+/+^ tumors since we clearly see the infiltration among the stroma as foci at the edge of the high grade PanIN (Figure 4) as observed in the PKI;POU2F3^+/+^ pancreatic tumors (Figure 2A).

Precisely, typical marker of macrophages such as F4/80 helped us to characterize the polarization types that are present narrow those specific structures. It appears that the alternative is the activated macrophages, also known as M2, and pro-tumoral spread around the PanIN of PKI;POU2F3^+/+^ tumors and not in PKI;POU2F3^−/−^. In the same way, enzymatic reaction shows the large contribution of neutrophils in PKI;POU2F3^+/+^ tumors only. Naphthol AS-D Chloro-Acetate Esterase (NCAE) has been proposed as a specific enzyme expressed in granulocytes lineage, but not in monocytes neither lymphocytes. The red color shows the presence of these cells in PKI;POU2F3^+/+^ tumors, once again around the high grade PanIN that are not observed in PKI;POU2F3^−/−^. This is confirmed by the gene expression analysis of CXCR2 (Figure 4) (2.86 ± 0.67 vs. 4.93 ± 0.57; *p*-value = 0.004) and chemokines responsible of their specific recruitment CXCL1 (Figure 4) (5.11 ± 2.01 vs. 13.30 ± 3.88; *p*-value = 0.003), CXCL2 (0.37 ± 0.24 vs. 0.77 ± 0.10; *p*-value = 0.05) and CXCL5 (37.40 ± 5.49 vs. 79.03 ± 36.50; *p*-value = 0.03) which are all down-regulated in the absence of POU2F3.

Taken together, our results described both Tumor-Associated Neutrophiles (TAN) and Macrophages (TAM), characterizing the type 2 immune response alongside the mast cells [40]. Knocking out POU2F3 avoids the recruitment of such cells, reduces the inflammatory response and thus, apoptosis which is much more activated in the presence of TUFT cells. The expression of pro-apoptotic factors such as Bax and Bid are significantly higher in PKI;POU2F3^+/+^ tumors than PKI;POU2F3^−/−^ (Figure 3; respectively 0.92 ± 0.35 vs. 1.75 ± 0.28; *p*-value = 0.09 and 3.86 ± 0,41 vs. 5.83 ± 1.11; *p*-value = 0.001). And in the opposite, mRNA of anti-apoptotic factor like Bcl-XL is more elevated in PKI;POU2F3^−/−^ tumors (Figure 4; 0.84 ± 0.17 vs. 0.22 ± 0.06; *p*-value = 0.001). PKI;POU2F3^+/+^ tumors show a pro-apoptotic equilibrium while it is neutralized in PKI;POU2F3^−/−^, reducing the call for immune cell income.

### 3.5. Systemic POU2F3 Deficiency Has Minimal Effect on Immune Cells Populations

POU2F3 is expressed in embryonic stem cells and the lineage related to the immune system such as thymocytes (CD4, CD8 and FoxP3 expressing cells), some antigen presenting cells (APC) like dendritic cells and also in granulocytes including neutrophils and mast cells [41]. We would assume that knocking out POU2F3 might have an effect on the global immune system development from the hematopoiesis that could explain the differences seen in the tumor adaptive response between PDAC of PKI;POU2F3^+/+^ and PKI;POU2F3^−/−^ mice.

No differences were observed in the spleen distribution regarding lymphoid and myeloid cells (Figure 5A) indicating that POU2F3 deletion does not impair hematopoiesis. We also screened cell populations harvested from the peritoneal cavity. We noticed in PKI;POU2F3^+/+^ as compare to PKI;POU2F3^−/−^ mice a slight decrease of the Large Peritoneal Macrophages (LPM) for the benefit of Small Peritoneal Macrophages (SPM) expressing PD-L2 (Figure 5B). The marker CD8a (T lymphocytes and DC cells) was found not significantly different between both PKI;POU2F3^+/+^ and PKI;POU2F3^−/−^ genotypes (2.83 ± 1.84 vs. 2.26 ± 0.66, *p*-value = 0.2) and neither does the CD64 marker (1.02 ± 0.06 vs. 1.33 ± 0.34, *p*-value = 0.02). Those data let us think that the adaptation of the immune response is probably triggered locally, on the tumor site, as a consequence of chemokine and/or cytokines production, such as IL13 instead of a direct effect on hematopoiesis.

## 4. Discussion

In this work we showed that POU2F3 deletion in PKI mice model inhibits the TUFT cells development without any effect on the PDAC development and growth. Nevertheless, it increases the expression of the epithelial-to-mesenchymal transition markers, decreases the inflammatory response and inhibits the liver metastasis formation.

Presently, it is clear that PDAC development is the consequence of the progressive accumulation of genetic mutations occurring at every steps of the disease [42] with the combination of epigenetic [35] and transcriptomic alterations [43]. During the last years, scientist focused their interest on the microenvironment of PDAC which is wide and complex, making difficult any attempt for drug delivery and showing the cell communications through the large spectra of factors that are released helping cancer to grow and spread [44,45,46,47,48]. Mostly, the microenvironment composition in PDAC is heterogeneous and variable between patients [43,49]. Therefore, understanding the influence of the microenvironment composition is important to gain further improvements in the immunotherapies as a personal treatment.

TUFT cells express high level of DCLK1 protein which serves as a marker [22]. These cells are well-known to regulate inflammation and immune cells recruitment in some tissues [23]. Development of TUFT cells is dependent of the expression of the POU2F3 transcriptional factor since POU2F3 inactivation results in the absence of TUFT cells [23,24]. Therefore, POU2F3 inactivation serves as a model of TUFT depletion. In the PDAC developed in PKI;POU2F3^−/−^ mice we found a significant decrease of the inflammatory response involving cells (mast cells, macrophages and granulocytes), cytokines (IL4, IL9 and IL13, IL17a, IL1b, IL10, IFNg, TNFa) chemokines (CXCL1, CXCL2) and their receptors (IL17AR, CXCR2). On the contrary, we found an increase in T lymphocytes as identified as CD8a+ and CD4+ cells. Interestingly, we also found that expression of the pro-apoptotic factors Bax and Bid were repressed in POU2F3^−/−^ mice, whereas the anti-apoptotic factor Bcl-XL was found overexpressed suggesting a lower apoptosis sensitivity in PDAC of POU2F3^−/−^ mice. Altogether, these data show that, under a POU2F3-deficiency background, inflammation is strongly reduced and apoptotic sensitivity decreased. Another interesting, although unexpected, thing revealed in this work was that expression of markers of endothelial-to-mesenchymal transition (VIM and CDH2) were found increased in the PDAC from PKI/POU2F3^−/−^ mice, whereas markers of epithelial cell differentiation (MUC, CDH1 and GATA6) were found decreased. Also, CD44 and the MMP19 molecules, which are involved in the metastatic process, were downregulated in POU2F3-deficient PDAC. These data indicate that POU2F3 inactivation, through a still unexplained molecular mechanism, influences the endothelial-to-mesenchymal transition. Despite every expectation, although endothelial-to-mesenchymal transition was stronger in PKI/POU2F3^−/−^ mice these animals presented less liver metastasis. Taken together, our work reveals that metastasis could be the direct consequences of the immune response driven by inflammation and/or apoptosis for the recruitment of T lymphocytes instead of the endothelial-to-mesenchymal transition process.

## 5. Conclusions

Since, POU2F3 expression may influences expansion of some components of the inflammatory and immune system we analyzed the cell populations in spleen and peritoneal cavity distribution by flow cytometry. Our results have shown that only minor differences were detected between POU2F3 deficient or proficient mice. This approach allowed us to show that the inflammatory and immune response observed in PDAC from POU2F3 deficient mice has been triggered directly from the tumor site and not through some alterations of the hematopoiesis. We hypothesize that TUFT cells, which are known to modulate the recruitment of inflammatory and immune cells, are at the origin of these differences.

## Figures and Tables

**Figure 1 biology-09-00341-f001:**
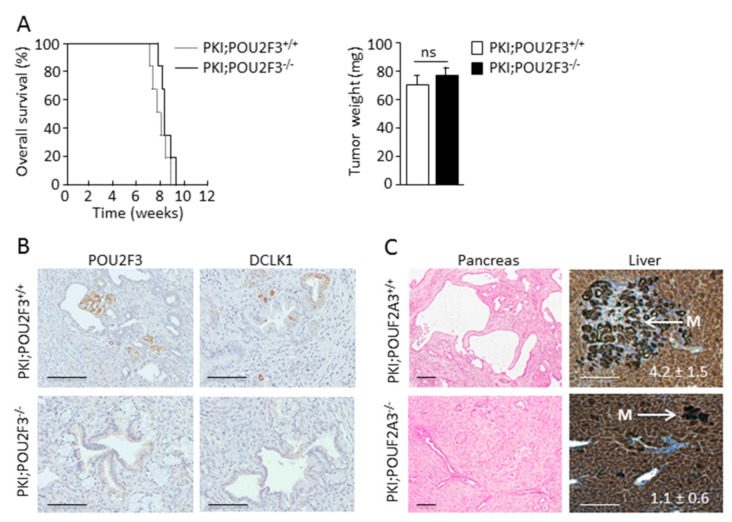
POU2F3 is not essential for pancreatic carcinogenesis but it is critical for liver metastasis. (**A**) Kaplan Meier survival curve of PKI;POU2F3^+/+^ and PKI;POU2F3^−/−^ mice (left panel) and the tumor weight at the sacrifice time (right panel). (**B**) Immuno-staining of POU2F3 and DCLK1 in the pancreas from PKI;POU2F3^+/+^ and PKI;POU2F3^−/−^ mice. (**C**) Hematoxylin and Eosin staining of the pancreas and liver from PKI;POU2F3^+/+^ and PKI;POU2F3^−/−^ mice. Values denote the metastasis per liver. Scale set as 100 µm.

**Figure 2 biology-09-00341-f002:**
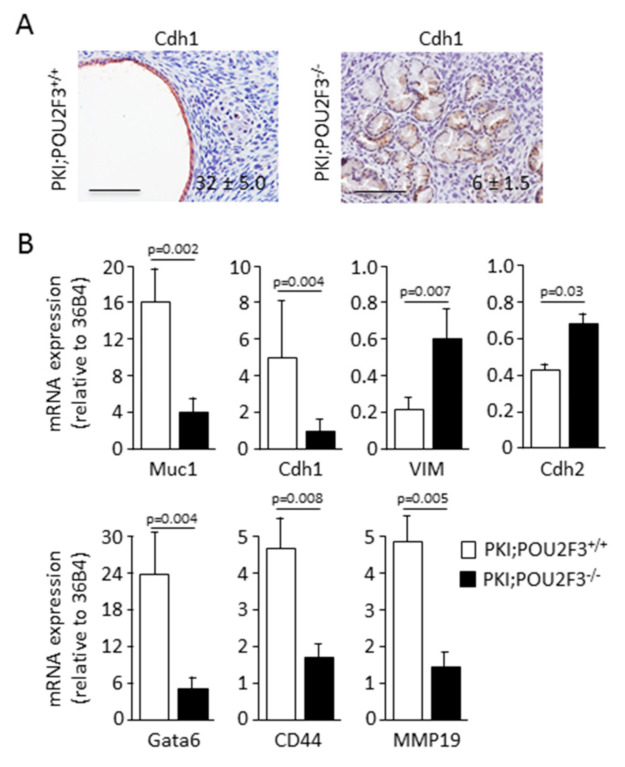
PDAC from PKI;POU2F3^−/−^ mice is more differentiated. (**A**) Immuno-staining of Cdh1 in pancreas from PKI;POU2F3^+/+^ and PKI;POU2F3^−/−^ mice. Values denote positive cells per field. (**B**) mRNA expression in pancreas from PKI;POU2F3^+/+^ and PKI;POU2F3 mice. Scale set as 100 µm.

**Figure 3 biology-09-00341-f003:**
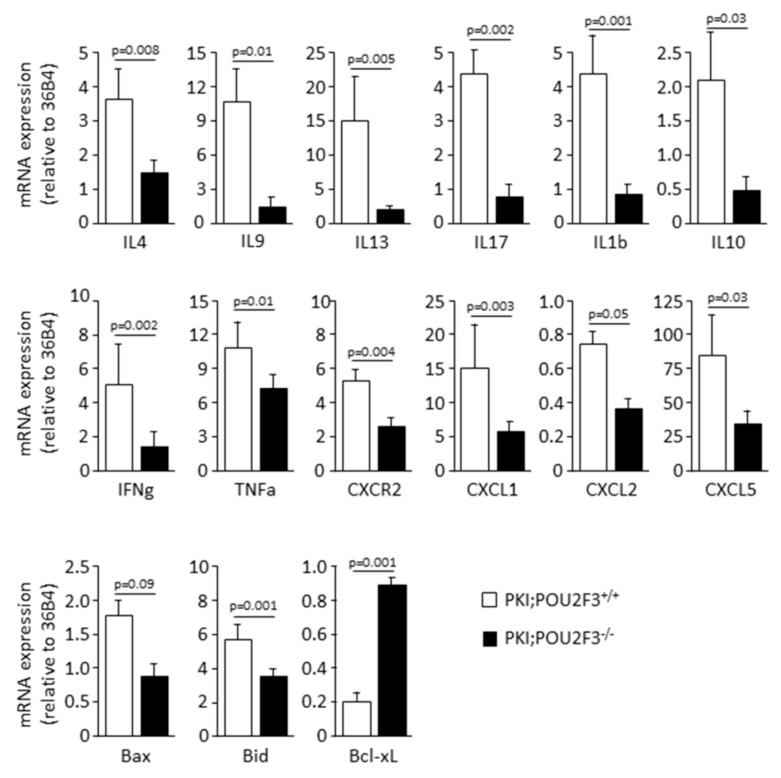
PDAC from PKI;POU2F3^−/−^ mice express less level of inflammatory markers and overexpress anti-apototic markers. mRNA expression in pancreas from PKI;POU2F3^+/+^ and PKI;POU2F3 mice.

**Figure 4 biology-09-00341-f004:**
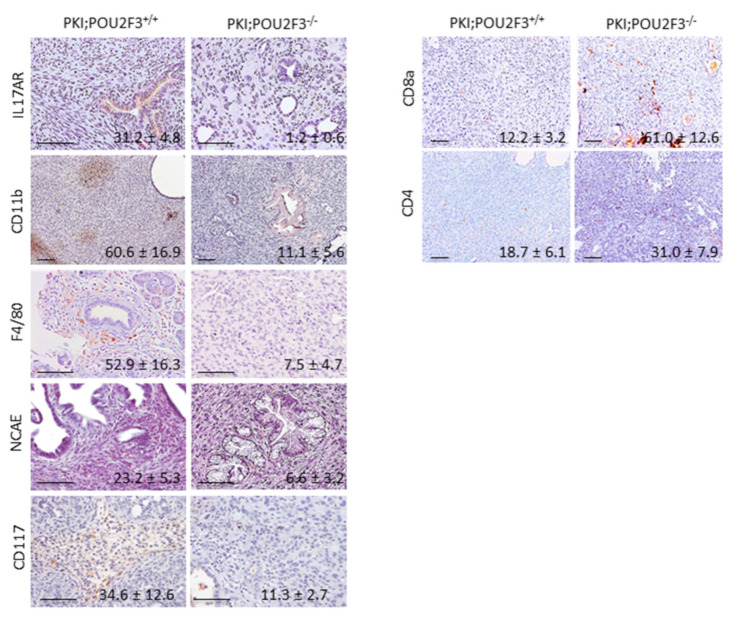
Immunological infiltration in PDAC. Immuno-staining of sections from pancreatic tumors. Values denote positive cells per field. Scale set as 100 µm.

**Figure 5 biology-09-00341-f005:**
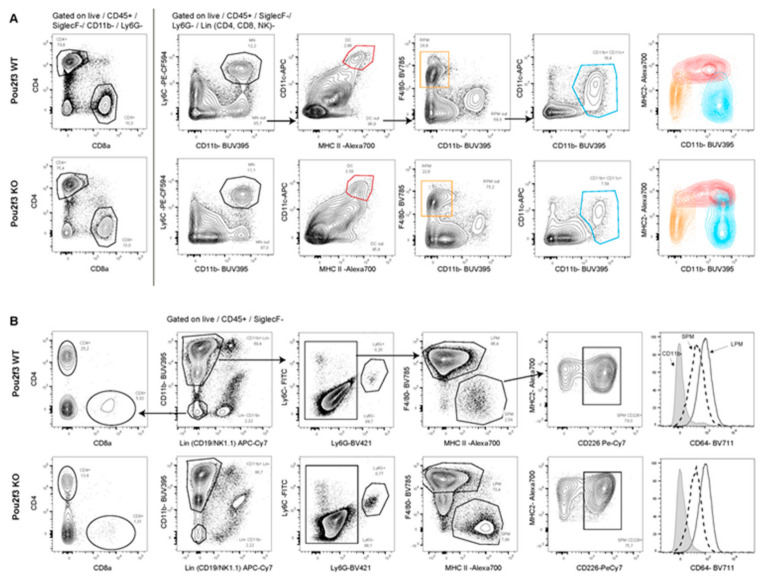
Immunophenotyping of cells from spleen and peritoneal cavity is similar in both genotypes. Cell sorting by flow cytometry of spleen cells (**A**) and from peritoneal cavity (**B**) from PKI;POU2F3^+/+^ and PKI;POU2F3^−/−^ mice.

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
