# Peer review of "Expression of POU2F3 Transcription Factor Control Inflammation, Immunological Recruitment and Metastasis of Pancreatic Cancer in Mice"

_biology, 2020, doi:10.3390/biology9100341_

Round 1
Reviewer 1 Report
This is an interesting manuscript that deals with the POU2F3 transcription factor modulates inflammation, immunological recruitment and metastasis of pancreatic cancer in mice
PDAC is one of the most lethal cancers worldwide and fourth leading cause of death by cancer in western countries. To understand the mechanisms of pancreatic cancer development and progression is still unknown.
TUFT cells have been shown as strong modulators of inflammatory cells in several tissues including pancreas. TUFT cells, also known as DCLK1+ cells, are dependent of the transcriptional factor POU2F3. Studies suggest that DCLK1+ cells in early stages of PDAC development such as ADM and PanIN in PDAC development through inflammation. The authors developed a mouse model (PDX1-Cre; KrasG12D;Ink4afl/fl known as PKI model, in which an aggressive PDAC is genetically induced, deficient or not for POU2F3. The authors are expected that deficiency of POU2F3 results in the absence of TUFT cells in PDAC. Notably, tumor growth is not significantly changed and development of liver metastasis was inhibited in POU2F3 deficient mice. Based on the interest authors trying to investigate the expression of POU2F3 transcription factor modulate inflammation, immunological recruitment and metastasis of pancreatic cancer in mice.
Figure 1 showed that TUFT cells are not essential for pancreatic carcinogenesis and POU2F3 inactivation decreases liver metastasis development.
Figure 2 demonstrated that POU2F3 inactivation enhances epithelial-to-mesenchymal transition in PDAC.
Figure 3 showed that POU2F3 inactivation enhances epithelial-to-mesenchymal transition in PDAC.
Figure 4 demonstrated that POU2F3 inactivation inhibits the recruitment inflammatory mediators into the PDAC.
Finally, Figure 5 showed that Systemic POU2F3 deficiency has minimal effect on immune cells populations.
Overall, this manuscript provides a comprehensive scientific evaluation of POU2F3 deletion in PKI mice model inhibits the TUFT cells development without any effect on PDAC development and growth. Nevertheless, it increases the expression of epithelial-to-mesenchymal transition markers, decreases the inflammatory response and inhibits the liver metastasis formation. Recent study showed that Pancreas-specific deletion of POU2F3 in KC mice (KPouC mice) resulted in a loss of tuft cells and accelerated tumorigenesis (DelGiorno KE etal Tuft Cells Inhibit Pancreatic Tumorigenesis in Mice by Producing Prostaglandin D2. Gastroenterology 2020 Jul 24;S0016-5085(20)34999-4). Current manuscript data look different than this recent publication. In addition, there are several contradictory results such as IL-4, IL-10, and IL-13, Bax, Bid and BclXL in this manuscript.
However, there are few of major concerns.
- Recent study showed that Pancreas-specific deletion of POU2F3 in KC mice (KPouC mice) resulted in a loss of tuft cells and accelerated tumorigenesis (DelGiorno KE etal Tuft Cells Inhibit Pancreatic Tumorigenesis in Mice by Producing Prostaglandin D2. Gastroenterology 2020 Jul 24;S0016-5085(20)34999-4). In your study suggest that deficiency of POU2F3 results in the absence of TUFT cells in PDAC and notably there is no significant change in the tumor growth. What is the reason? Which factor is responsible for no change in the tumor growth?
- In Figure 4, IL-4, IL-10, and IL-13 are anti-inflammatory cytokines. Why PKI;POU2F3-/- pancreas reduced the IL-4, IL-10, and IL-13 mRNA expression? In addition, an anti-apoptotic Bcl-XL mRNA is more elevated in PKI;POU2F3-/- tumors. Why?
- What about serum metastasis marker CA125 in PKI;POU2F3-/- mice?
- The marker CD8a (T 239 lymphocytes and DC cells) was found not significantly different between both PKI;POU2F3+/+ and 240 PKI;POU2F3-/- genotypes (2.83 ± 1.84 vs. 2.26 ± 0.66, p-value = 0.2) and neither does the CD64 marker (1.02 ± 0.06 vs. 1.33 ± 0.34, p-value = 0.02). What is the reason?
Author Response
Reviewer 1
Comments and Suggestions for Authors
This is an interesting manuscript that deals with the POU2F3 transcription factor modulates inflammation, immunological recruitment and metastasis of pancreatic cancer in mice
PDAC is one of the most lethal cancers worldwide and fourth leading cause of death by cancer in western countries. To understand the mechanisms of pancreatic cancer development and progression is still unknown.
TUFT cells have been shown as strong modulators of inflammatory cells in several tissues including pancreas. TUFT cells, also known as DCLK1+ cells, are dependent of the transcriptional factor POU2F3. Studies suggest that DCLK1+ cells in early stages of PDAC development such as ADM and PanIN in PDAC development through inflammation. The authors developed a mouse model (PDX1-Cre; KrasG12D;Ink4afl/fl known as PKI model, in which an aggressive PDAC is genetically induced, deficient or not for POU2F3. The authors are expected that deficiency of POU2F3 results in the absence of TUFT cells in PDAC. Notably, tumor growth is not significantly changed and development of liver metastasis was inhibited in POU2F3 deficient mice. Based on the interest authors trying to investigate the expression of POU2F3 transcription factor modulate inflammation, immunological recruitment and metastasis of pancreatic cancer in mice.
Figure 1 showed that TUFT cells are not essential for pancreatic carcinogenesis and POU2F3 inactivation decreases liver metastasis development.
Figure 2 demonstrated that POU2F3 inactivation enhances epithelial-to-mesenchymal transition in PDAC.
Figure 3 showed that POU2F3 inactivation enhances epithelial-to-mesenchymal transition in PDAC.
Figure 4 demonstrated that POU2F3 inactivation inhibits the recruitment inflammatory mediators into the PDAC.
Finally, Figure 5 showed that Systemic POU2F3 deficiency has minimal effect on immune cells populations.
Overall, this manuscript provides a comprehensive scientific evaluation of POU2F3 deletion in PKI mice model inhibits the TUFT cells development without any effect on PDAC development and growth. Nevertheless, it increases the expression of epithelial-to-mesenchymal transition markers, decreases the inflammatory response and inhibits the liver metastasis formation. Recent study showed that Pancreas-specific deletion of POU2F3 in KC mice (KPouC mice) resulted in a loss of tuft cells and accelerated tumorigenesis (DelGiorno KE etal Tuft Cells Inhibit Pancreatic Tumorigenesis in Mice by Producing Prostaglandin D2. Gastroenterology 2020 Jul 24;S0016-5085(20)34999-4). Current manuscript data look different than this recent publication. In addition, there are several contradictory results such as IL-4, IL-10, and IL-13, Bax, Bid and BclXL in this manuscript.
However, there are few of major concerns.
Recent study showed that Pancreas-specific deletion of POU2F3 in KC mice (KPouC mice) resulted in a loss of tuft cells and accelerated tumorigenesis (DelGiorno KE etal Tuft Cells Inhibit Pancreatic Tumorigenesis in Mice by Producing Prostaglandin D2. Gastroenterology 2020 Jul 24;S0016-5085(20)34999-4). In your study suggest that deficiency of POU2F3 results in the absence of TUFT cells in PDAC and notably there is no significant change in the tumor growth. What is the reason? Which factor is responsible for no change in the tumor growth?
We thank this reviewer for this important remark. DelGiorno and collaborators (2020) performed their studies on LSL-KrasG12D/+; Ptf1aCre/+ mice which induces a less aggressive pancreatic cancer compared with the Pdx1-cre ; LSL-KrasG12D ; Ink4a/Arffl/fl animal model in which the tumor kill mice around 10 weeks old. In this model cell transformation is evident after 1 week old within any additional treatment, such as induction of inflammation by injection of cerulein, as utilized in that paper. This induced inflammation was essential to promote PDAC in these mice.
Our PDAC mice model is a genetic model without any additional needs for promoting cancer. In our experimental models we observed not changes in tumor growth as consequence of absence of TUFT cells. However, we observed that TUFT cells are driving a specific immune response promoting and accelerating the inflammation and then leading to the evasion of metastasis. This is why we hypothesize that TUFT cells have not direct influence in transformation, at least in our animal model, but it strongly influences the tumor microenvironment and consequently its capacity to develop metastasis.
In summary, we assume that TUFT cells do not influence directly cell transformation or cell growth but it is able to influences tumor microenvironment.
In Figure 4, IL-4, IL-10, and IL-13 are anti-inflammatory cytokines. Why PKI;POU2F3-/- pancreas reduced the IL-4, IL-10, and IL-13 mRNA expression? In addition, an anti-apoptotic Bcl-XL mRNA is more elevated in PKI;POU2F3-/- tumors. Why?
This is an interesting remark. As it has been shown by Gerbe et al., (2016) TUFT cells express IL4, IL10 and IL13 for recruiting specific immune cells and therefore, misleading the immune response for facing adequately the tumor growth. As expected, in the absence of TUFT cells, we found a strong decrease of cytokines expression related to the type 2 immune response.
Regulation of apoptosis by BCL2-related proteins is complex and the function of one is conditioned by the expression of the others since they form complexes. In this case, the high rate of the anti-apoptotic Bcl-XL is associated to the low levels of the pro-apoptotic factors Bax and Bid suggesting lower apoptosis sensitivity in PDAC of POU2F3-/- mice. Because we didn’t found any effect of TUFT cells with the transformation process we hypothesize about a link between inflammation and apoptosis control.
What about serum metastasis marker CA125 in PKI;POU2F3-/- mice?
We thanks for this interesting remark. However, we consider measure of the metastatic markers is out of the scope of the work at this stage of the project.
The marker CD8a (T lymphocytes and DC cells) was found not significantly different between both PKI;POU2F3+/+ and PKI;POU2F3-/- genotypes (2.83 ± 1.84 vs. 2.26 ± 0.66, p-value = 0.2) and neither does the CD64 marker (1.02 ± 0.06 vs. 1.33 ± 0.34, p-value = 0.02). What is the reason?
The immunological response in the cancer field is complex. As remarked by this reviewer, any differences were found on the amount of these cells indicating that TUFT cells are regulating a specific immunological way which seems to be essential for liver metastasis development. This pathway does not involve CD8a+ and CD64+ cells. We are currently trying to define this important way.

Reviewer 2 Report
In the manuscript ‘Expression of POU2F3 transcription factor control inflammation, immunological recruitment and metastasis of pancreatic cancer in mice’, Bintz et al., assess pancreatic cancer progression and inflammation in a mouse model of PDAC with and without tuft cells. They show that, although there is no difference in survival between the two cohorts, tumors from POU2F3 knockout mice have higher expression of EMT markers. The authors suggest that knockout mice have higher metastatic load and lower inflammatory cell infiltration. While this subject is interesting and important to the field, the authors have incompletely evaluated their data. Further, the manuscript requires further proofreading.
Concerns:
Abstract – The authors state that liver metastasis was ‘almost inhibited’, where they seem to have meant ‘almost completely inhibited’.
3.1:
Line 115 – ‘Figure 1A’ is actually 1B. An insert is necessary to show that Pou2f3 staining is nuclear.
Line 117 – Please quantify Pou3f3 and Dclk1 immunohistochemistry
Line 123 – ‘Figure 1B’ is actually 1A in the figure
Line 124 – ‘Figure 1B’ is actually 1A in the figure – reorder the text or figure
- A control is missing. You need to show that Pou2f3 ablation has no effect on normal development of the pancreas. A developmental defect could explain phenotypic differences.
- DelGiorno et al., 2020 (Gastroenterology) has now shown that Pou2f3 is required for tuft cell formation in the pancreas and that Dclk1 is not tuft cell-specific.
3.2:
Line 131 – Only H&E is shown. This (or at least the representative photo chosen) is not sufficient to demonstrate metastatic load. The image chosen shows necrosis; the cells suggested to be metastases could be cancer cells or immune cells. Please conduct IHC for a marker that will demonstrate that these cells are metastatic cancer cells (Krt19, Pdx1). Stain multiple sections from each mouse at a defined distance apart (i.e. 200um); quantify IHC.
3.3:
- These data are interesting and suggest more EMT markers in Pou2f3-/- mice as compared to control, but are contrary to the lack of metastases described in section 3.2. These data are more consistent with recent publications (DelGiorno et al., 2020 Gastroenterology; Hoffman et al., 2020 Cell Mol Gastroenterol Hepatol) demonstrating that tuft cells are anti-tumorigenic. I recommend having a pathologist review both the primary tumors and the metastases to get a more complete description of the aggressiveness/differentiation of tumors from each genotype. Please add to the discussion why your results are in opposition to what was shown in these papers.
Line 158 – Please insert a reference that shows MMP19 and CD44 to be pro-metastatic
-Please quantify Cdh1 IHC.
3.4:
Line 175 – The data described are for Figure 3, not 4, please correct
Line 184 – this is figure 4, not figure 3
Line 188 – This reference shows that tuft cells secrete IL-25, not IL-13.
Line 205 – This is figure 4, not 3
Line 223 – This is figure 3, not 4
-Please add bar graphs showing the distribution and significance of IHC quantification in figure 4
- The result that Pou2f3+/+ tumors have more markers of apoptosis is interesting, but it is the opposite of what was shown in DelGiorno et al., 2020 (Figure S3). Please conduct IHC for CC3 on Pou2f3+/+ and Pou2f3-/- tumors and quantify.
Author Response
Reviewer 2
Comments and Suggestions for Authors
In the manuscript ‘Expression of POU2F3 transcription factor control inflammation, immunological recruitment and metastasis of pancreatic cancer in mice’, Bintz et al., assess pancreatic cancer progression and inflammation in a mouse model of PDAC with and without tuft cells. They show that, although there is no difference in survival between the two cohorts, tumors from POU2F3 knockout mice have higher expression of EMT markers. The authors suggest that knockout mice have higher metastatic load and lower inflammatory cell infiltration. While this subject is interesting and important to the field, the authors have incompletely evaluated their data. Further, the manuscript requires further proofreading.
Concerns:
Abstract – The authors state that liver metastasis was ‘almost inhibited’, where they seem to have meant ‘almost completely inhibited’.
This sentence was corrected accordingly
3.1:
Line 115 – ‘Figure 1A’ is actually 1B. An insert is necessary to show that Pou2f3 staining is nuclear.
This mistake was corrected.
Line 117 – Please quantify Pou3f3 and Dclk1 immunohistochemistry
Pou3f3 and Dclk1 staining was measured and data was included in the results section of the revised version of the manuscript.
Line 123 – ‘Figure 1B’ is actually 1A in the figure
This mistake was corrected.
Line 124 – ‘Figure 1B’ is actually 1A in the figure – reorder the text or figure
As suggested by this reviewer the order was modified in the text.
A control is missing. You need to show that Pou2f3 ablation has no effect on normal development of the pancreas. A developmental defect could explain phenotypic differences.
Thanks to this reviewer for this important observation. We performed histological comparisons, at the birth time, between Pou2f3+/+ and Pou2f3-/- mice pancreas and not evident alterations were found. Also, we measured pancreatic level of amylase (as a marker of exocrine development) and glycaemia (as a marker of endocrine function) and no differences were observed. Finally no inflammatory signs were found in the pancreas. We assume that inactivation of Pou2f3 have not obvious effects of pancreas development.
DelGiorno et al., 2020 (Gastroenterology) has now shown that Pou2f3 is required for tuft cell formation in the pancreas and that Dclk1 is not tuft cell-specific.
We agree with this remark, but unfortunately these both Pou2f3 and Dclk1 are the better tools available for the moment to study the role of TUFT cells in PDAC.
3.2:
Line 131 – Only H&E is shown. This (or at least the representative photo chosen) is not sufficient to demonstrate metastatic load. The image chosen shows necrosis; the cells suggested to be metastases could be cancer cells or immune cells. Please conduct IHC for a marker that will demonstrate that these cells are metastatic cancer cells (Krt19, Pdx1). Stain multiple sections from each mouse at a defined distance apart (i.e. 200um); quantify IHC.
Thanks to this reviewer for her/his comment. In the revised version of the manuscript we included a more evident picture of the liver metastasis. Quantification was done on these types of lesions.
3.3:
- These data are interesting and suggest more EMT markers in Pou2f3-/- mice as compared to control, but are contrary to the lack of metastases described in section 3.2. These data are more consistent with recent publications (DelGiorno et al., 2020 Gastroenterology; Hoffman et al., 2020 Cell Mol Gastroenterol Hepatol) demonstrating that tuft cells are anti-tumorigenic. I recommend having a pathologist review both the primary tumors and the metastases to get a more complete description of the aggressiveness/differentiation of tumors from each genotype. Please add to the discussion why your results are in opposition to what was shown in these papers.
Thanks to this reviewer for her/his comment. Pictures were revised by our pathologist and he didn’t found morphological differences in PDAC between both PKI;POU2F3+/+ and PKI;POU2F3-/- pancreas genotypes. The apparent contradiction between our work and DelGiorno paper may be certainly explained by the animal model utilized.
Line 158 – Please insert a reference that shows MMP19 and CD44 to be pro-metastatic
References were included
-Please quantify Cdh1 IHC.
Data was quantified and included in Figure 2.
3.4:
Line 175 – The data described are for Figure 3, not 4, please correct
This mistake was corrected
Line 184 – this is figure 4, not figure 3
This mistake was corrected
Line 188 – This reference shows that tuft cells secrete IL-25, not IL-13.
This mistake was corrected. In fact “secreted by TUFT cells” was replaced by “stimulated by TUFT cells”.
Line 205 – This is figure 4, not 3
This mistake was corrected
Line 223 – This is figure 3, not 4
This mistake was corrected
-Please add bar graphs showing the distribution and significance of IHC quantification in figure 4
This point was modified in Figure 4.
- The result that Pou2f3+/+ tumors have more markers of apoptosis is interesting, but it is the opposite of what was shown in DelGiorno et al., 2020 (Figure S3). Please conduct IHC for CC3 on Pou2f3+/+ and Pou2f3-/- tumors and quantify.
Thanks to this reviewer for her/his comment. As said above, differences that we found with DelGiorno paper may be due to the strong different animal models of PDAC utilized in both stuides.

Reviewer 3 Report
To the authors
Jennifer Bintzand colleagues report about POU2F3 as a key factor and new therapeutic target for control PDAC progression, reviewing the current shreds of evidence that pinpoint metastatic process as a phenomenon associated to a higher expression of influencer of the tumour-stroma cross talk affecting the tumoral microenvironment and the different strategies discussing the available PDAC mouse models. The manuscript is well written, nonetheless, there are few sections that deserve to be restructured, in order to achieve the level and comprehensive overview that a journal like Cancers would aim to.
Major points to consider in subsequent versions:
RT-qPCR: The authors show mRNA expression levels relative to controls. For consistency reasons, absolute values should be added and mRNA expression levels in healthy (or healthy representing) controls should be included.
Immunophenotyping: did the authors employ unstained/isotype control stained controls?
Statistics: the authors employed Kaplan-Meuer equation. Kaplan-Meier curve-is often used to assist readers of a paper in the interpretation. However, mistakes and distortions frequently arise in the display and interpretation of survival plots n the Kaplan-Meier method, the follow-up time is divided into intervals, with limits corresponding to the follow-up time between events, with or without censorship. The likelihood of participants at the beginning of each interval to develop the event by the end of each interval is estimated. Survival at the end of each interval equals the product of cumulative survivals to the end of the previous interval by conditional survival in that interval. Individuals censored in one interval no longer count as individuals at risk in the next interval. Can the authors comment on this topic?
Moreover, Mann-Whitney tests were employed. I would assume that no normal distribution of the data could be reached. This should be highlighted as a study limitation and discussed appropriately.
Mice model: when discussing the methodology applied for number selection it is not clear for my understanding, how the author selected the sufficient experimental setting. Indeed, in the in vivo experiments, the sample size has been should be calculated in a rigorous way, i.e. by using G*Power software (power of for example 80% and 0.05 statistical level, etc.). Assuming an effect-size of for example, 0.4 with statistical significance of α <;0.05 and a power of 80%. Can the authors comment on this topic?
Introduction/discussion. Because of the intimate interactions, the capacity of PDAC cells, as well as other tumour cells, and CAF as can be also discussed, since several examples have been recently published, pointing out CAF role in mediating drug resistance (i.e. PMID: 30866547). Moreover, in the last paragraph (ref 42), the authors partially mentioned in the manuscript that the tumour milieu stimulate crucial immune effectors. I think there are now good indications to think that protein inhibitors are not always that specific and/or selective and that such a capability might open-up a therapeutic window, especially in light of the surprising finding pinpointed by the authors. Nonetheless, from a preclinical standpoint, several cancers with terribly poor prognosis could benefit from novel insights derived from these data in light of novel findings in humans regarding lower inflammatory microenvironment or less reactive one in human PDAC model (i.e. PMID: 31277479). The manuscript translational relevance would benefit from this introduction/discussion expansion.
Author Response
Reviewer 3
Comments and Suggestions for Authors
Jennifer Bintz and colleagues report about POU2F3 as a key factor and new therapeutic target for control PDAC progression, reviewing the current shreds of evidence that pinpoint metastatic process as a phenomenon associated to a higher expression of influencer of the tumour-stroma cross talk affecting the tumoral microenvironment and the different strategies discussing the available PDAC mouse models. The manuscript is well written, nonetheless, there are few sections that deserve to be restructured, in order to achieve the level and comprehensive overview that a journal like Cancers would aim to.
Major points to consider in subsequent versions:
RT-qPCR: The authors show mRNA expression levels relative to controls. For consistency reasons, absolute values should be added and mRNA expression levels in healthy (or healthy representing) controls should be included.
Thanks to the reviewer for her/his comment. The goal of our work was to compare the effect of TUFT cells in PDAC development and production of liver metastasis. This is why we only used both PKI;POU2F3+/+ and PKI;POU2F3-/- pancreas. We consider that other genotypes are unrelated to our goals.
Immunophenotyping: did the authors employ unstained/isotype control stained controls?
The controls showed in the figures were slides stained with the same antibody but from the mouse model expressing or not POU2F3. We also used some slides stained with the secondary antibody in absence of the primary antibody for checking any potential background, with no significant signal.
Statistics: the authors employed Kaplan-Meier equation. Kaplan-Meier curve-is often used to assist readers of a paper in the interpretation. However, mistakes and distortions frequently arise in the display and interpretation of survival plots n the Kaplan-Meier method, the follow-up time is divided into intervals, with limits corresponding to the follow-up time between events, with or without censorship. The likelihood of participants at the beginning of each interval to develop the event by the end of each interval is estimated. Survival at the end of each interval equals the product of cumulative survivals to the end of the previous interval by conditional survival in that interval. Individuals censored in one interval no longer count as individuals at risk in the next interval. Can the authors comment on this topic?
We agree with the view of this reviewer but we utilized the Kaplan-Meier method for analyzing the survival time of the animal because it is the most popular and easily interpretable by the scientist and readers.
Moreover, Mann-Whitney tests were employed. I would assume that no normal distribution of the data could be reached. This should be highlighted as a study limitation and discussed appropriately.
The Mann-Whitney tests are employed in this case because distribution of data is symmetric, proceed from distinct populations and samples do not affect each other. There are some other statistical methods which can be used to analyze our data but this seems appropriate to us.
Mice model: when discussing the methodology applied for number selection it is not clear for my understanding, how the author selected the sufficient experimental setting. Indeed, in the in vivo experiments, the sample size has been should be calculated in a rigorous way, i.e. by using G*Power software (power of for example 80% and 0.05 statistical level, etc.). Assuming an effect-size of for example, 0.4 with statistical significance of α <;0.05 and a power of 80%. Can the authors comment on this topic?
Thanks to this reviewer for this important remark. The animal model which we used is the mixture of 4 independent genotypes which is very tedious and long to be obtained. We used 6 animals in each group (PKI;POU2F3+/+ and PKI;POU2F3-/-) and found no differences at all in their survival time. In our opinion, the statistical analysis performed in this study showed enough robustness (8 ± 0.88 vs 9 ± 0.22 weeks for PKI;POU2F3+/+ and PKI;POU2F3-/- respectively).
Introduction/discussion. Because of the intimate interactions, the capacity of PDAC cells, as well as other tumour cells, and CAF as can be also discussed, since several examples have been recently published, pointing out CAF role in mediating drug resistance (i.e. PMID: 30866547). Moreover, in the last paragraph (ref 42), the authors partially mentioned in the manuscript that the tumour milieu stimulate crucial immune effectors. I think there are now good indications to think that protein inhibitors are not always that specific and/or selective and that such a capability might open-up a therapeutic window, especially in light of the surprising finding pinpointed by the authors. Nonetheless, from a preclinical standpoint, several cancers with terribly poor prognosis could benefit from novel insights derived from these data in light of novel findings in humans regarding lower inflammatory microenvironment or less reactive one in human PDAC model (i.e. PMID: 31277479). The manuscript translational relevance would benefit from this introduction/discussion expansion.
We also thanks to this reviewer for this important remark. We modified the discussion section accordingly to the reviewer comments and included both suggested referees in the revised version of the manuscript.

Round 2
Reviewer 1 Report
The revised manuscript provides a comprehensive scientific evaluation of POU2F3 deletion in PKI mice model inhibits the TUFT cells development without any effect on PDAC development and growth. Nevertheless, it increases the expression of epithelial-to-mesenchymal transition markers, decreases the inflammatory response and inhibits the liver metastasis formation.
Overall, the revised manuscript has improved and authors answered the questions.
Reviewer 3 Report
The authors have clarified several of the questions I raised in my previous review. Most of the major problems have been addressed by this revision. No further comments from this reviewer.